# Adherence and Dietary Composition during Intermittent vs. Continuous Calorie Restriction: Follow-Up Data from a Randomized Controlled Trial in Adults with Overweight or Obesity

**DOI:** 10.3390/nu13041195

**Published:** 2021-04-05

**Authors:** Sarah T. Pannen, Sandra González Maldonado, Tobias Nonnenmacher, Solomon A. Sowah, Laura F. Gruner, Cora Watzinger, Karin Nischwitz, Cornelia M. Ulrich, Rudolf Kaaks, Ruth Schübel, Mirja Grafetstätter, Tilman Kühn

**Affiliations:** 1Division of Cancer Epidemiology, German Cancer Research Center (DKFZ), 69120 Heidelberg, Germany; s.gonzalezmaldonado@dkfz-heidelberg.de (S.G.M.); t.nonnenmacher@dkfz-heidelberg.de (T.N.); s.sowah@dkfz-heidelberg.de (S.A.S.); l.gruner@dkfz-heidelberg.de (L.F.G.); Cora.Watzinger@med.uni-muenchen.de (C.W.); Karin.Nischwitz@med.uni-heidelberg.de (K.N.); r.kaaks@Dkfz-Heidelberg.de (R.K.); ruth.schuebel@gmx.de (R.S.); m.grafetstaetter@mailbox.org (M.G.); 2Department of Diagnostic and Interventional Radiology, University Hospital Heidelberg, 69120 Heidelberg, Germany; 3Medical Faculty, Heidelberg University, 69120 Heidelberg, Germany; 4Huntsman Cancer Institute and Department of Population Health Sciences, University of Utah, Salt Lake City, UT 84112, USA; neli@hci.utah.edu; 5Institute for Global Food Security, Queen’s University Belfast, Belfast BT9 5DL, UK; 6Heidelberg Institute of Global Health, Heidelberg University, 69120 Heidelberg, Germany

**Keywords:** obesity, weight loss, fasting, intermittent energy restriction, compliance, energy intake, diet quality, food records

## Abstract

Although intermittent calorie restriction (ICR) has become popular as an alternative weight loss strategy to continuous calorie restriction (CCR), there is insufficient evidence on diet quality during ICR and on its feasibility over longer time periods. Thus, we compared dietary composition and adherence between ICR and CCR in a follow-up analysis of a randomized trial. A total of 98 participants with overweight or obesity [BMI (kg/m^2^) 25–39.9, 35–65 years, 49% females] were randomly assigned to ICR, operationalized as a “5:2 diet” (energy intake: ~100% on five non-restricted (NR) days, ~25% on two restricted (R) days), or CCR (daily energy intake: ~80%). The trial included a 12-week (wk) intervention phase, and follow-up assessments at wk24, wk50 and wk102. Apart from a higher proportion of energy intake from protein with ICR vs. CCR during the intervention (wk2: *p* < 0.001; wk12: *p* = 0.002), there were no significant differences with respect to changes in dietary composition over time between the groups, while overall adherence to the interventions appeared to be good. No significant difference between ICR and CCR regarding weight change at wk102 was observed (*p* = 0.63). However, self-reported adherence was worse for ICR than CCR, with 71.1% vs. 32.5% of the participants reporting not to or only rarely have followed the regimen to which they were assigned between wk50 and wk102. These results indicate that within a weight management setting, ICR and CCR were equivalent in achieving modest weight loss over two years while affecting dietary composition in a comparable manner.

## 1. Introduction

The increasing prevalence of overweight and obesity worldwide indicates the need for feasible weight reduction strategies [1]. The dietary concept of intermittent calorie restriction (ICR), which consists of periods of calorie restriction and regular calorie intake, has continued to gain popularity over the past decade not only as an alternative weight loss strategy to the conventional concept of continuous calorie restriction (CCR) [2,3], but also for its putative impact on various health outcomes and longevity [4,5]. Given that difficulties with adhering to standard CCR diets are well known [6], it was hypothesized that ICR concepts could have the potential for better adherence rates, as they require a focus on calorie restriction only for defined time periods and could therefore offer better long-term practicability in everyday life compared to CCR [2,7,8]. Several clinical trials have reported equivalent weight reduction with ICR as with CCR in the short term [9,10,11,12] and over the course of one year [13,14,15,16]. However, the current literature investigating subjective satisfaction with ICR as well as long-term adherence, efficacy and safety in humans is limited [17]. Concomitantly, there are concerns about the occurrence of compensatory overeating during ICR, as seen in animal models [18]. Also, little is known about the nutrient and food composition of ICR diets [10,19,20]. A previous study that analyzed nutritional composition during ICR and CCR showed that ICR was related to unfavorable food choices with regard to the intake of fruits and berries, vegetables, fiber, vitamin C and sugar compared to CCR, indicating the need for further research on this topic [20].

We previously published results from the HELENA Trial, a 50 wk randomized controlled clinical trial on the effects of ICR, operationalized as the “5:2 diet”, and CCR on metabolic biomarkers and body weight [13]. Here, we carried out extended analyses using data of the HELENA Trial on the feasibility of ICR compared to CCR. Our specific aim was to investigate whether there are differences in dietary composition on the levels of nutrients and foods during ICR vs. CCR over the course of the clinical trial. In addition, we aimed to evaluate participants’ self-reports on adherence as well as changes in body weight between baseline and a final follow-up interview 102 wks after baseline.

## 2. Materials and Methods

### 2.1. Study Design and Subjects

For this analysis, data from the HELENA Trial (Healthy nutrition and energy restriction as cancer prevention strategies: a randomized controlled intervention trial), a three-armed, randomized controlled open-label trial carried out to evaluate the metabolic effects of ICR, operationalized as the “5:2 diet”, compared to CCR over one year, were used for exploratory post hoc analyses. The HELENA Trial was registered at clinicaltrials.gov (NCT02449148) and approved by the Ethics Committee of the Medical Faculty of Heidelberg. The trial was powered to detect differences between the trial arms in changes in the expression of 82 candidate genes in adipose tissue (primary endpoint) and in pre-defined established metabolic, anthropometric, and body composition parameters (secondary endpoints). A detailed description of the rationale, design, sample size calculation and main outcomes of the study has been published previously [13,21].

A total of 150 non-smoking non-diabetic females and males with overweight or obesity (50% females) between 35 and 65 years (BMI range in kg/m^2^: 25–39.9) were randomly assigned (1:1:1) to ICR, CCR or to a control group. The web-based software RANDI2 with a block size of six, stratified by age and sex was used for randomization [22]. Participants were enrolled over the course of one year via flyer and poster campaigns as well as word of mouth (May 2015–May 2016). All participants gave written informed consent to participate in the study. For our present analyses on dietary composition and adherence we only used data from participants in the ICR (*n* = 49) and the CCR (*n* = 49) group, as our goal was to compare ICR and CCR as two alternative weight loss regimens. The participants allocated to the control group (*n* = 52) were only included for analyses on weight development at the 2-y follow-up, as our previous analyses had shown a moderate initial amount of weight loss among the controls [13].

The HELENA Trial included a 12-wk intervention phase (wk0–wk12), a 12-wk maintenance phase (wk12–wk24), and a 26-wk follow-up phase (wk24–wk50). At wk102, a telephone follow-up interview was carried out during which participants self-reported their current body weight and their dietary behavior during the year after the end of the study. Anthropometry measures were taken by study personnel at baseline (wk0), wks 12, 24 and 50. During the intervention phase (wks 4, 6, 8, 10), participants were interviewed by telephone to report their body weight. Participants received monetary incentives for participation in the study after completion of the intervention phase (wk12) and at the final study visit (wk50).

Details on side effects of the dietary interventions have been described in detail before [13]. In brief, five participants in the ICR group reported mild physical impairments on restricted (R) days, and three reported mild physical impairments on non-restricted (NR) days during the first 12 weeks. In the CCR group, one participant reported mild cognitive impairments, and another participant reported mild physical impairments during the first 12 weeks. Otherwise, no adverse effects were reported over the two-year duration of the study.

### 2.2. Dietary Interventions

Trained dietitians undertook nutrition counselling sessions at baseline, and after two weeks. Motivation to maintain the prescribed diet was provided in the following biweekly phone calls (wks 4, 6, 8, 10) and at the end of the guided intervention phase (wk12). No further recommendations and motivation were given thereafter. Participants in all three study groups were instructed to follow the principles of a “healthy balanced diet” (55% energy from carbohydrates, 15% from protein, and 30% from fat) according to the recommendations of the German Nutrition Society (Deutsche Gesellschaft für Ernährung—DGE) [23,24]. They were encouraged to follow a diet high in complex carbohydrates and fiber, low in high-fat foods, refined sugar and sweetened beverages and to prefer low-fat dairy alternatives. Dietary schedules of both calorie restriction groups (ICR and CCR) were arranged to have an equal net mean energy restriction of ~20% of their isoenergetic energy requirement per week, while the participants in the control group were instructed not to perform any calorie restriction. The individual isoenergetic energy requirement of each participant was estimated at baseline, by multiplying the resting energy expenditure (calculated with the gender-specific Harris-Benedict equation [25]) by the Physical Activity Level (PAL) factor (determined using questionnaire data on physical activity and profession).

Participants in the ICR group were advised to follow a “5:2 diet” with an energy intake of 25% of the isoenergetic energy requirement on two self-selected, non-consecutive R days and an isoenergetic “healthy balanced diet” (100% energy intake) on the remaining five NR days (net weekly energy intake of approximately 80%). For R days, a meal plan list with pre-selected food items was provided on the basis of which participants were free to choose their meals (four vegetable items, two low-fat dairy products, one item out of each of the meat/fish, carbohydrate, and fruits group). CCR participants were encouraged to reduce energy intake to ~80% of the individual energy requirement daily. Based on the 7-day food records filled out at baseline, personalized diet plans incorporating individual eating habits were provided by the dietitians.

### 2.3. Dietary Assessment

Dietary intake was assessed using paper-based self-reported 7-day food records filled in at baseline (wk0), during the second (wk2) and last week of the intervention phase (wk12), and at the end of the follow-up phase (wk50). Participants were instructed to provide detailed information on the type, portion size and preparation methods of foods and ingredients of mixed dishes and recipes consumed. Digital kitchen scales and a picture book containing various types of foods in different portion sizes were provided to all participants to simplify the estimation of the amount of foods consumed. Food records were reviewed by a dietitian for completeness. Analyzing the 7-day food records, the study staff transcribed hand-written details on meal intakes into PRODI 6.8 (Nutri-Science GmbH, Hausach, Germany), a dietary software based on the German Nutrient Data Base (Bundeslebensmittelschlüssel, version 3.02), to obtain nutrient and food intakes. All recorded foods and beverages were added by the day of intake. An overview of energy, nutrient, and food intake was displayed and spot-checked. For each of the four time points, the intake of all seven protocol days were included to determine the mean energy, nutrient and food intake per day. For the ICR group, the mean intake was additionally calculated separately for NR and R days. Further details on the calculation of the mean intakes in the ICR and the CCR group are given in the Appendix A.

### 2.4. Statistical Analyses

Baseline characteristics were summarized using descriptive statistics (mean ± SD for continuous variables; percentages for categorical variables). Changes over time are shown as mean log relative differences (%) ± standard error log relative differences (%). All continuous variables were checked for normality by calculating the Shapiro-Wilk test. While differences between groups at one time point were assessed by Fisher’s exact test (2-Tail) (for categorical variables) or by ANOVA F-test (for continuous variables), within-group differences were assessed using a paired t-test. As many nutrient and food intake variables were not normally distributed but right-skewed, they were log2-transformed for ANOVA and t-test analyses (see Appendix A for an example of the crude and log2-transformed data distribution in the food group “beef, veal, pork and mutton”). Analysis on differences in body weight, energy, nutrient and food intake across the study groups over time were conducted using a linear mixed model for repeated measures with baseline values as the reference, conducted as intention-to-treat analysis. Mixed models were adjusted for age and sex, and *p*-values were obtained for time-by-treatment interactions. All data analyses were undertaken using SAS, version 9.4 (SAS Institute Inc., Cary, NC, USA).

## 3. Results

### 3.1. Baseline Characteristics of the Study Population

Baseline characteristics of the participants are shown in Table 1. There were 49 participants (49% females) in each group. Participants had a mean age of 49.4 (±9.0) years and 50.5 (±8.0) years in the ICR and the CCR group, respectively.

Among the 98 participants included in the study in the ICR and the CCR group at baseline, respectively, 47 and 46 participants completed the intervention phase (wk12), 47 and 45 the maintenance phase (wk24), and 45 and 41 the follow-up phase (wk50). The response rate to the telephone follow-up in wk102 was 91.8% (*n* = 45) in the ICR group and 81.6% (*n* = 40) in the CCR group. Information on the reasons for withdrawal can be found in the CONSORT diagram on the HELENA Trial (Appendix A). At baseline, food records were filled out by all study participants. Completion rates for dietary records at later time points were as follows; wk2, during the intervention: 45 participants in the ICR (91.8%) and 46 participants in the CCR group (93.9%); wk12, last week of the intervention: 47 in the ICR (95.9%; R day data available of 44 participants) and 46 in the CCR group (93.9%); wk50, follow-up: 41 in the ICR (83.7%; R day data available of 12 participants) and 38 in the CCR group (77.6%).

### 3.2. Adherence to the Dietary Interventions

#### 3.2.1. Energy Intake and Body Weight

Participants in both intervention groups reduced their energy intakes strongly during the intervention phase (wk2, wk12) and moderately at the end of the follow-up phase of the active study (wk50), with no significant between-group differences at any time (Table 2). The recommended net weekly energy intake in the ICR and CCR group was 80% of the isoenergetic energy requirement, whereas the reported mean intakes at wk2 accounted for 56.0% and 61.2% (wk12: 55.3%, 61.3%; wk50: 65.1%, 70.2%) of the isoenergetic energy requirement (Appendix A), indicating significantly lower reported mean energy intakes in both groups compared to the calculated prescribed energy intakes (*p* < 0.001). Notably, the reported energy intakes at baseline were already −30.4% ± 5.3% (ICR) and ‑25.2% ± 3.1% (CCR) lower than the estimated mean isoenergetic energy requirement (*p* < 0.001). Thus, compared to the reported baseline energy intakes, the mean reported energy intakes in the ICR and CCR group at wk2 were 79.4% and 79.6% (wk12: 78.1%, 78.6%; wk50: 90.4%, 91.7%, Appendix A). The predicted weight loss according to the reported energy deficit in relation to the isoenergetic energy requirement during the 12-wk intervention phase (ICR: −8.2 ± 3.9 kg; CCR: −6.7 ± 2.3 kg) was higher than the actual measured weight loss in the ICR (−6.5 ± 4.8 kg) and the CCR group (−4.7 ± 3.5 kg) (Appendix A) [26].

Among participants in the ICR group, the mean reported energy intakes on NR and R days at wk2 were 67.8% and 26.9% (wk12: 65.3%, 26.9%; wk50: 67.9%, 21.6%) respectively compared to the isoenergetic energy requirement, and 94.4% and 42.0% (wk12: 92.2%, 40.4%; wk50: 94.3%, 35.6%) respectively compared to the baseline energy intake. Thus, rather than the planned energy intake of 100% of the estimated isoenergetic energy requirement on NR days and 25% on R days, participants in the ICR group consumed less energy than prescribed on NR days (*p* < 0.001, Appendix A).

Overall, weight loss was similar in the ICR and the CCR group from baseline to the 2-year follow-up assessment at wk102 (Figure 1). Importantly, weight at wk102 was assessed by telephone interview. However, weight values at wks 4 to 10, which were also based on self-reports via telephone, were plausible compared to measured values at wks 0 and 12, indicating accuracy of self-reported weight at wk102 (Appendix A). Within both intervention groups, weight loss compared to baseline values was significant at all times (Appendix A). The tendency towards a greater weight loss in the ICR compared to the CCR participants during the intervention phase (*p* = 0.053) is in line with the slightly higher relative decreases in the reported energy intake from wk0 to wk12 (*p* = 0.15, Appendix A). Compared to body weight at wk50 (ICR: −5.2% ± 1.2%, CCR: −4.9% ± 1.1%), there was a slight tendency for weight regain in the year after the end of the active study (between wk50 and wk102) in the ICR group (Figure 1). 

However, weight loss compared to baseline at wk102 did not differ between groups (ICR: −4.3% ± 1.0%, −5.0% ± 1.1%, *p* = 0.63). Compared to participants in the control group (weight reduction of −1.8% ± 0.9%), two years after baseline, relative weight loss was still significantly higher in both intervention groups (Control vs. ICR: *p* = 0.048; Control vs. CCR: *p* = 0.02).

#### 3.2.2. Self-Reported Adherence

Analysis on self-reported frequency of performed R days in the ICR group over the course of one year was previously published by Schübel et al. [13]. The main result is cited here again in order to facilitate an overall assessment of adherence to the intervention. While adherence to the 2 R days per week was high during the intervention phase (mean days at ∼25% energy intake = 1.8 d/wk), “the number of participants who performed 2 energy-restricted days per week decreased across the maintenance and follow-up phases, from 15 (32.6%) in week 24 to 9 (21.4%) in week 50” [13].

When asked whether they had continued with the diets initially assigned to them between the end of the active study (wk50) and the telephone follow-up (wk102), 71.1% of the participants in the ICR group reported that they had not at all or only rarely done so, compared to 32.5% in the CCR group (*p* < 0.001, Appendix A). In the ICR group, weight maintenance among participants, who reported a continuation with the diet after the end of the active study (wk50), was significantly higher (−5.7% ± 1.2%) compared to those who did not continue (−1.0% ± 1.5%) (*p* = 0.025) (Appendix A). The mean number of weeks during which participants in the ICR group reported to have followed the dietary regimen assigned to them was 15.8 ± 13.2 wks. Due to the low number of participants, who reported to have not all continued with the CCR diet (*n* = 2), no similar analysis on weight maintenance according to continuation could be carried out (Appendix A). As opposed to the high rates of self-reported continuation with ICR and CCR in the year after the active study (wk50), a vast majority of participants in both groups (ICR: 80.0%, CCR: 82.5%) also reported that they had re-adopted dietary habits detrimental to weight maintenance they had before the start of the study (Appendix A). When asked whether the participants had followed a diet other than the ICR (“5:2 diet”) and CCR approach during the year after the end of the active study (wk50–wk102), 24.4% of the participants in the ICR, and 10.0% of the participants in the CCR group reported that they had done so (Appendix A). The types of diets reported varied among individuals and included (but were not limited to): “Low carb” diet approaches, “Weight Watchers diet program”, and “16:8 ICR diet” (Appendix A).

Slightly less than half of the participants in the ICR (46.7%) and the CCR group (45.0%) stated that they continued adhering to the recommendations for a “healthy balanced diet” in agreement with the recommendations of the DGE in the year after completion of the study. There was non-significant tendency towards better weight maintenance among participants who reported to have continued following a “healthy balanced diet” in the ICR group at wk102 (−6.1% ± 1.6% vs. −2.8% ± 1.2%, *p* = 0.10), while no such tendency was observed in the CCR group (−5.6% ± 1.8% vs. −4.4% ± 1.5%, *p* = 0.51, Appendix A).

### 3.3. Dietary Composition

Changes in macronutrient composition during the intervention were comparable between the ICR and the CCR group, with an overall increase in energy% (E%) from protein, carbohydrates and fiber, accompanied by a reduction in E% from fat, which was in line with the recommended macronutrient composition in the HELENA Trial (Table 3 and Appendix A). While E% from protein increased more with ICR than with CCR during the intervention phase (wk2: *p* < 0.001; wk12: *p* = 0.002), there were no significant differences between the groups in terms of E% from fat, carbohydrates and fiber (Table 3 and Appendix A) as well as in the absolute intakes of all displayed macronutrients (Appendix A). The only difference in micronutrient intake between the groups was a higher intake of β-carotene in the ICR compared to the CCR group at wk2, even though it cannot be ruled out that this difference, which was only observed at one time point, was a chance finding given the high number of tests (Appendix A).

On the food group level, there was a tendency towards higher consumptions of fruits and fruit products (wk2: *p* < 0.001, wk12: *p* = 0.014) grains, grain products and rice (wk12: *p* = 0.003), and fats and oils (wk2: *p* = 0.015), accompanied by a lower consumption of sweets, sugar and ice cream (wk12: *p* = 0.010) in the CCR compared to the ICR group (Table 4). However, none of these differences reached statistical significance between the groups (time-treatment interactions: *p* > 0.05). Differences between the ICR and CCR group were significant regarding changes in the intake of bread (baseline to wk2, baseline to wk12) as well as sausage and processed meat (baseline to wk12). Of note, these differences in changes over time were due to different baseline levels, considering that differences in intakes at wk2 and wk12 were not statistically significant (*p* > 0.05). Overall, both groups adapted their diet according to the prescribed DGE recommendations for a “healthy balanced diet” in a comparable manner, as reflected by higher fiber and vegetable intake and lower intake of fat, pastry and sweets compared to baseline.

In the ICR group, in accordance with the recommendations on meal components rich in protein and vegetables for R days, dietary composition on R compared to NR days was characterized by higher proportions of E% from protein and fiber (Appendix A). Thus, the differences in E% from protein between ICR and CCR reported above may have been due to higher E% from protein on R days in the ICR group. Despite the overall lower energy intake on R days, the intake of β-carotene (Appendix A) and the consumption of vegetables and vegetable products (Appendix A) was greater on R than on NR days. While carbohydrates accounted for a comparable proportion on both, R and NR days, E% from fat was lower on R days than on NR days (Appendix A).

## 4. Discussion

In this randomized trial, weight loss from baseline to a follow-up assessment after two years was similar with ICR (−4.3% ± 1.0%) and CCR (−5.0% ± 1.1%). A higher proportion of people in the ICR (71.1%) than in the CCR group (32.5%) reported that they had not or only rarely followed the respective diet after the active study between wk50 and wk102. With regard to dietary composition, ICR and CCR affected nutrient and food intake in a comparable manner. The only exception was E% from protein, which increased significantly more with ICR compared with CCR during the intervention phase (wk2, wk12), due to the high E% from protein on R days, as defined by the study protocol. This difference was no longer significant after follow-up at wk50. With regard to the energy intakes on NR and R days in the ICR group, the reported intakes suggest that the intended energy difference of 75% between NR and R days was not fully achieved, while there was no overcompensation of calorie intake on NR days (mean energy intake on NR days: ~90–95%, R days: ~40%).

The similar weight loss with both regimens over one year is in line with other published studies on weight loss with ICR and CCR [14,15,27,28,29,30] and indicates that in the short-term, ICR seems to be as easy to follow as CCR. At wk 102, participants in the ICR group reported to have integrated R days in only 16 out of the previous 52 weeks, i.e., in the year after the initial one-year study. Weight change at wk102 after baseline was slightly, but not significantly lower with ICR than CCR (−4.3% ± 1.0% vs. −5.0% ± 1.1%). Again, this result is in agreement with those from two Australian trials on “5:2 ICR”, from which data on weight after 24 months were reported [27,28]. The first study showed −3.5 kg weight loss with ICR vs. −4.5 kg with CCR among people with metabolically healthy overweight [27], the second study among people with type 2 diabetes showed a weight loss of −3.9 kg with both regimens [28].

The findings on weight loss two years after baseline from our study and the two Australian studies mentioned above do not support the notion that the “5:2 diet”, a mild type of ICR, is more easy to follow over longer durations compared to CCR. By contrast, similar to a previous study by Harvie et al. [12], our study suggests that more people assigned to CCR may be willing to continue with this diet beyond a controlled trial situation compared to ICR. This was also reflected by the observation that more people in the ICR group started following alternative diets in the year after the study compared to those in the CCR group. Interestingly, however, a subgroup analysis among participants of our trial in the ICR group suggested that those who reported that they had continued with ICR in the year after the trial, were more successful in maintaining weight than those who had not. Thus, ICR may be a long-term alternative for weight control for some people, which is in line with previous weight loss interventions showing that adherence to specific lifestyle modifications and long-term weight maintenance may greatly vary between individuals [31,32,33].

It has been suggested that ICR may lead to compensatory overeating on NR days, as seen in animal studies [17] and among humans [34,35]. Contrary to these concerns, but consistent with the present findings, spontaneous moderate reductions in energy intake on NR days were observed in human ICR trials based on a “5:2 diet” [11,12,19,36] or on alternate-day fasting, another popular form of ICR [14,37]. In the HELENA Trial, regardless of whether energy intake on NR and R days was referenced to the calculated individual isoenergetic energy requirement or to the reported baseline intake, the overall intended energy difference of 75% between NR and R days was not reached on average. Thus, there may have been a slight shift from the prescribed ICR diet to a CCR diet, and the total weekly energy restriction in the ICR group may have been achieved not only by calorie restriction on R days, but also by a moderate spontaneous reduction on NR days.

Overall, the ICR and the CCR group adapted their dietary composition according to the prescribed macronutrient composition and the DGE guidelines for a “healthy balanced diet” in a comparable manner, with both groups being most adherent at the beginning of the study. Although fiber intake was increased, participants following ICR and CCR still did not meet the DGE recommendations of a fiber intake of 30 g/day [38]. In a study by Sundfør et al., which had a very similar study design as the HELENA Trial [15], more favorable changes in dietary composition were observed in the CCR compared to the ICR group with regard to the intake of fruits and berries, vegetables, fiber, sugar and vitamin C [20]. Even though a similar tendency for higher fruit consumption with CCR than ICR participants was observed in the HELENA Trial, apart from a significantly higher intake of E% from protein in the ICR than in the CCR group, both interventions affected dietary composition in a comparable manner. High consumption of protein during ICR has been reported in previous studies [14,19,30]. This seems plausible, as a higher protein intake may improve satiety [39], especially on R days, as reflected by the higher actual intake of E% from protein on R days in our study. No indication for a less favorable change in micronutrient intake in the ICR compared to the CCR group, as reported by Sundfør et al. [20], was observed in our study. Thus, ICR may not necessarily imply a suboptimal micronutrient supply, and recommendations for a balanced dietary pattern as provided in the HELENA Trial may help to prevent micronutrient deficiencies during ICR.

A key strength of the current study was the comprehensive analysis of dietary composition during ICR, which was assessed at four different time points over one year. The open-label and “non-feeding” concept of the study may limit its internal validity to some degree. However, the present analyses do suggest that the protocol of the study worked to a large extent, and the “real-life” situation of the study participants lends the findings from the HELENA Trial a greater external validity than a feeding trial. In this context, the relatively low drop-out rate should be mentioned as a strength of the HELENA Trial as well. It was a limitation of the present analysis that apart from weight loss (wk0–wk50), all tools used for the assessment of adherence to the dietary intervention during the study were based on self-reported data. Although participants were asked whether they had followed a dietary intervention during the year after the end of the active study, it cannot be ruled out that body weight was influenced by other types of interventions, such as pharmaceutical products, that were not specifically assessed during the follow-up interview. Although PRODI 6.8 software was based on the most recent version of the German Nutrient Data Base, methodological limitations inherent to any nutrient database (varying nutrient contents, effects of preparation etc.) must be acknowledged. As the average reported baseline energy intakes in both groups were already about one quarter below the isoenergetic energy requirement, it is highly likely that underreporting, which is known to be a common issue in the assessment of self-reported dietary intakes, particularly among people with overweight [40], was present in the HELENA Trial. However, the level of underreporting in the ICR and the CCR group appears to be similar across all time points, given that relative decreases in reported energy intake throughout the intervention and observed relative decreases in body weight were proportional.

In summary, the results of this study indicate that ICR and CCR may be equivalent approaches for weight management in people with overweight or obesity over longer periods of time without evidence of superiority of one approach in terms of diet quality. While adherence to ICR and CCR seems to be similar on average, there was variation in achieved weight loss between study participants in each trial arm. In the long-term, more people seem to have difficulty following ICR than CCR. Future, pooled analyses are needed to find out whether specific determinants for successful weight loss with either method exist.

## Figures and Tables

**Figure 1 nutrients-13-01195-f001:**
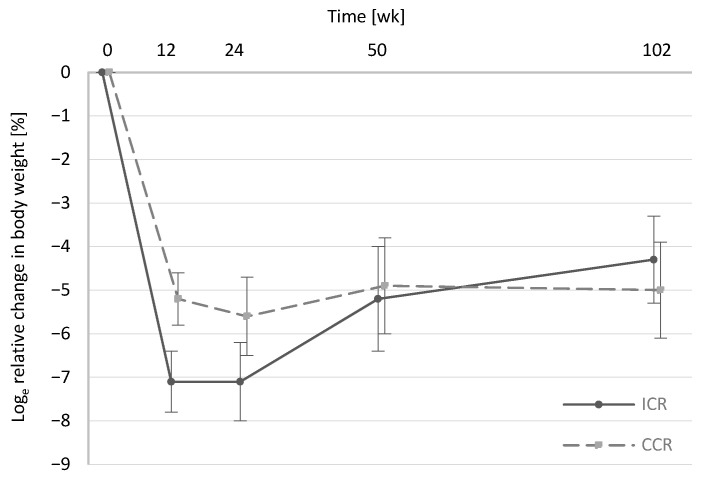
Relative changes in body weight by study group from baseline to wk102. Data are shown as means of log_e_ percentage changes ± standard error of log percentage changes, with baseline values as the reference for ICR (*n* = 49) and CCR (*n* = 49). Intention-to-treat statistical analyses were performed using a linear mixed model adjusted for age and sex. There were no significant differences (*p* > 0.05) between the intervention groups. Results on anthropometric parameters including body weight at baseline (wk0) and wk12, wk24 and wk50 have been published in detail before [13]. Body weight from these time points is shown here again to facilitate a better interpretation of body weight at the 2-year follow-up assessment (wk102). CCR, continuous calorie restriction; ICR, intermittent calorie restriction.

**Table 1 nutrients-13-01195-t001:** Baseline characteristics of the study population ^1^.

	ICR *n* = 49; 49% Females	CCR *n* = 49; 49% Females
Age (years)	49.4 ± 9.0	50.5 ± 8.0
Weight (kg)	96.4 ± 15.8	92.5 ± 15.7
BMI (kg/m^2^)	32.0 ± 3.8	31.2 ± 4.0
Waist (cm)	104.7 ± 12.3	103.7 ± 11.9
Professional degree ^2^		
Higher education entrance qualification	31 (62.3)	28 (59.6)
Primary school	7 (14.3)	5 (10.6)
Secondary school certificate	11 (22.5)	14 (29.8)
Blood pressure		
Systolic (mmHg)	139.4 ± 18.7	136.0 ± 16.7
Diastolic (mmHg)	87.2 ± 9.9	87.3 ± 8.7
Glucose (mg/dL)	92.7 ± 7.5	93.9 ± 7.5
HbA1_c_ (%)	5.4 ± 0.3	5.5 ± 0.4
HDL cholesterol (mg/dL)	54.1 ± 14.4	56.2 ± 16.3
LDL cholesterol(mg/dL)	124.5 ± 22.4	122.5 ± 31.5
Triglycerides (mg/dL)	130.0 ± 83.8	121.2 ± 66.3

^1^ Values are means ± SDs for continuous variables and counts (% of answers) for categorical variables unless otherwise indicated; *n* = 98. CCR, continuous calorie restriction; ICR, intermittent calorie restriction. ^2^ Two participants did not report their educational level.

**Table 2 nutrients-13-01195-t002:** Estimated isoenergetic energy requirement, calculated prescribed energy intakes according to the intervention and reported energy intakes at wk0, wk2, wk12 and wk50 ^1^.

	Energy Requirement (kcal/d) ^2^	Reported Intake wk0 (kcal/d)	Prescribed Intake (kcal/d) ^3^	Reported Intake wk2 (kcal/d)	Change wk0–wk2 (%) ^1^	*p*-Value ^4^	Reported Intake wk12 (kcal/d)	Change wk0–wk12 (%) ^1^	*p*-Value ^4^	Reported Intake wk50 (kcal/d)	Change wk0–wk50 (%) ^1^	*p*-Value ^4^
ICR (daily)	2630.8 ± 490.9	2053.3 ± 746.0	2067.1 ± 385.7	1469.9 ± 454.5	−30.2 ± 5.3 *	<0.001	1438.4 ± 486.1	−32.1 ± 5.2 *	<0.001	1689.8 ± 624.4	−16.2 ± 5.4 *	<0.001
CCR (daily)	2507.3 ± 378.5	1981.0 ± 476.5	1970.1 ± 297.3	1537.4 ± 342.1	−25.4 ± 3.4 *	<0.001	1529.4 ± 364.8	−25.8 ± 2.8 *	<0.001	1768.8 ± 553.2	−14.2 ± 5.0 *	<0.001
*p*-value ^5^					0.23			0.15			0.51	
ICR NR day ^6^	-	-	2630.8 ± 490.9	1783.0 ± 641.4	-	<0.001	1699.9 ± 568.3	-	<0.001	1765.9 ± 633.6	-	<0.001
ICR R day ^6^	-	-	657.7 ± 122.7	706.0 ± 220.6	-	0.45	700.7 ± 246.5	-	0.74	566.8 ± 161.7	-	0.09

^1^ Values are means ± SDs unless otherwise indicated. Changes are compared to baseline values and are shown as means of log_e_ percentage changes ± standard error of log percentage changes (%). * Significant within group change over time compared to baseline at *p* < 0.05 from paired *t*-test. Data were included from 98 participants (wk0 *n* = 49 for ICR and CCR; wk2 *n* = 45 for ICR, *n* = 46 for CCR; wk12 *n* = 47 for ICR (R days: *n* = 44), *n* = 46 for CCR; wk50 *n* = 41 for ICR (R days: *n* = 12), *n* = 38 for CCR). CCR, continuous calorie restriction; ICR, intermittent calorie restriction; NR, non-restricted; R, restricted. ^2^ Isoenergetic energy requirement was calculated by the Harris-Benedict equation multiplied with the PAL. Isoenergetic energy requirement at wk12 and wk50 is reported in the Online Appendix A. ^3^ Calculated prescribed energy intake for ICR and CCR = ~80% of calculated isoenergetic energy requirement; ICR: Calculated prescribed energy intake on NR days = ~100% of calculated isoenergetic energy requirement, calculated prescribed energy intake on R days = ~25% of calculated isoenergetic energy requirement. ^4^
*p*-values for differences between prescribed and reported energy intake were calculated by paired *t*-test. ^5^
*p*-values for time-treatment interactions were calculated with linear mixed models adjusted for age and sex (ICR vs. CCR). ^6^ Mean NR and R days weighed by the reported number of days that each participant conducted R days.

**Table 3 nutrients-13-01195-t003:** Changes in relative macronutrient intake by intervention group ^1^.

		wk0	wk2	Change wk0–wk2 (%) ^1^	wk12	Change Wk0–wk12 (%) ^1^	wk50	Change wk0–wk50 (%) ^1^
Protein (E%)	ICR	15.8 ± 2.5	20.4 ± 3.3	25.7 ± 3.1 *	19.6 ± 4.0	20.9 ± 3.6 *	17.1 ± 3.0	9.3 ± 2.7 *
	CCR	15.6 ± 3.2	17.7 ± 2.7	11.7 ± 3.3 *	17.2 ± 2.6	8.5 ± 2.9 *	16.3 ± 2.5	3.3 ± 4.0
	*p*-value ^2,3^	0.83 ^2^	<0.001 ^2^	<0.001 ^3^	<0.001 ^2^	0.002 ^3^	0.20 ^2^	0.26 ^3^
Fat (E%)	ICR	38.1 ± 5.6	29.4 ± 5.9	−26.7 ± 3.4 *	30.6 ± 6.3	−22.6 ± 3.4 *	34.0 ± 6.3	−10.7 ± 3.8 *
	CCR	37.2 ± 5.2	29.5 ± 5.6	−23.2 ± 3.3 *	31.5 ± 6.5	−17.1 ± 3.6 *	32.9 ± 8.2	−13.0 ± 5.4 *
	*p*-value ^2,3^	0.39 ^2^	0.89 ^2^	0.48 ^3^	0.51 ^2^	0.27 ^3^	0.50 ^2^	0.89 ^3^
SFA (E%)	ICR	15.3 ± 3.1	11.8 ± 2.9	−24.8 ± 4.0 *	11.4 ± 3.0	−30.0 ± 4.1 *	13.0 ± 3.4	−14.7 ± 4.7 *
	CCR	14.3 ± 2.9	10.8 ± 2.8	−29.0 ± 3.7 *	10.9 ± 2.8	−28.8 ± 4.9 *	12.5 ± 4.1	−17.3 ± 6.4 *
	*p*-value ^2,3^	0.12 ^2^	0.08 ^2^	0.73 ^3^	0.41 ^2^	0.62 ^3^	0.55 ^2^	0.83 ^3^
MUFA (E%)	ICR	11.0 ± 2.8	9.5 ± 2.2	−14.0 ± 4.3 *	9.2 ± 3.0	−20.7 ± 6.3 *	9.9 ± 2.7	−9.4 ± 5.5 *
	CCR	11.0 ± 2.5	10.1 ± 2.5	−8.6 ± 4.4	9.8 ± 2.3	−11.6 ± 4.7	10.4 ± 3.5	−8.0 ± 6.5
	*p*-value ^2,3^	0.91 ^2^	0.24 ^2^	0.34 ^3^	0.28 ^2^	0.41 ^3^	0.50 ^2^	0.60 ^3^
PUFA (E%)	ICR	4.9 ± 1.7	4.6 ± 1.3	−5.5 ± 5.2	4.7 ± 1.6	−5.2 ± 6.3	4.9 ± 2.1	−0.8 ± 6.3
	CCR	4.9 ± 1.6	5.5 ± 2.1	9.9 ± 7.0	5.1 ± 1.6	5.6 ± 5.0	4.4 ± 1.3	−2.8 ± 6.1
	*p*-value ^2,3^	0.87 ^2^	0.027 ^2^	0.10 ^3^	0.28 ^2^	0.37 ^3^	0.21 ^2^	0.38 ^3^
Carbohydrates (E%)	ICR	44.4 ± 6.4	47.0 ± 5.5	6.0 ± 2.7 *	46.9 ± 6.9	5.3 ± 2.4	46.4 ± 6.7	2.6 ± 2.8
	CCR	45.4 ± 6.5	49.7 ± 5.3	9.5 ± 2.3 *	48.5 ± 5.7	7.1 ± 2.9 *	48.3 ± 8.2	5.1 ± 4.0
	*p*-value ^2,3^	0.43 ^2^	0.021 ^2^	0.27 ^3^	0.23 ^2^	0.69 ^3^	0.26 ^2^	0.65 ^3^
Fiber (g/d) ^4^	ICR	17.9 ± 8.1	21.8 ± 7.4	25.1 ± 6.4 *	18.9 ± 7.9	8.4 ± 7.2	19.8 ± 8.9	12.0 ± 7.4
	CCR	17.5 ± 6.1	24.4 ± 7.7	33.6 ± 4.9 *	21.8 ± 9.1	22.3 ± 5.0 *	21.9 ± 12.4	19.3 ± 6.9 *
	*p*-value ^2,3^	0.79 ^2^	0.11 ^2^	0.07 ^3^	0.11 ^2^	0.08 ^3^	0.39 ^2^	0.31 ^3^

^1^ Values are means ± SDs unless otherwise indicated. Changes are compared to baseline values and are shown as means of log_e_ percentage changes ± standard error of log percentage changes (%). ^*^ Significant within group change over time compared to baseline at *p* < 0.05 from paired t-test. Data were included from 98 participants (wk0 *n* = 49 for ICR and CCR; wk2 *n* = 45 for ICR, *n* = 46 for CCR; wk12 *n* = 47 for ICR (R days: *n* = 44), *n* = 46 for CCR; wk50 *n* = 41 for ICR (R days: *n* = 12), *n* = 38 for CCR). CCR, continuous calorie restriction; ICR, intermittent calorie restriction. SFA, Saturated fatty acids; MUFA, Monounsaturated fatty acids; PUFA, Polyunsaturated fatty acids. ^2^
*p*-values for differences between groups at each time point were calculated by ANOVA *F*-test. ^3^
*p*-values for time-treatment interactions were calculated with linear mixed models adjusted for age and sex. ^4^ The ratio of insoluble to soluble fiber was approximately 2:1 in both groups and was constant over time.

**Table 4 nutrients-13-01195-t004:** Changes in the intake of foods by intervention group ^1^.

	wk0	wk2	Change wk0–wk2 (%) ^1^	wk12	Change wk0–wk12 (%) ^1^	wk50	Change wk0–wk50 (%) ^1^
Vegetables and vegetable products (g/d)							
ICR	135.1 ± 93.8	283.2 ± 133.4	80.9 ± 12.4 *	282.0 ± 207.7	63.1 ± 13.2 *	189.1 ± 123.5	33.6 ± 11.9 *
CCR	154.3 ± 89.7	254.4 ± 118.8	56.5 ± 10.9 *	234.2 ± 162.7	42.3 ± 10.8 *	204.2 ± 136.0	28.7 ± 13.2 *
*p*-value ^2,3^	0.30 ^2^	0.28 ^2^	0.13 ^3^	0.22 ^2^	0.11 ^3^	0.60 ^2^	0.76 ^3^
Fruits and fruit products (g/d)							
ICR	180.6 ± 219.1	172.3 ± 86.8	18.1 ± 12.3	168.5 ± 96.1	10.0 ± 17.5	176.0 ± 151.0	5.3 ± 15.8
CCR	195.8 ± 144.0	258.8 ± 138.0	50.5 ± 16.3 *	225.7 ± 123.2	36.3 ± 14.5 *	245.1 ± 175.4	40.5 ± 18.5 *
*p*-value ^2,3^	0.68 ^2^	<0.001 ^2^	0.08 ^3^	0.014 ^2^	0.27 ^3^	0.06 ^2^	0.18 ^3^
Bread (g/d)							
ICR	126.9 ± 68.5	101.8 ± 61.9	−22.0 ± 9.7 *	98.1 ± 53.7	−22.3 ± 9.0 *	114.5 ± 74.6	−12.6 ± 7.6
CCR	97.5 ± 50.0	102.8 ± 46.9	9.9 ± 8.4	107.1 ± 55.0	12.1 ± 9.9	102.8 ± 50.0	−0.8 ± 10.2
*p*-value ^2,3^	0.017 ^2^	0.94 ^2^	0.023 ^3^	0.43 ^2^	0.002 ^3^	0.42 ^2^	0.27 ^3^
Grains and grain products, rice (g/d)							
ICR	20.9 ± 22.2	36.0 ± 31.6	39.0 ± 19.5	22.3 ± 20.4	7.9 ± 16.4	24.6 ± 24.9	−7.8 ± 19.8
CCR	31.9 ± 32.3	51.7 ± 44.1	46.1 ± 14.4 *	39.1 ± 31.8	37.1 ± 17.2 *	27.6 ± 26.5	−1.2 ± 21.4
*p*-value ^2,3^	0.05 ^2^	0.06 ^2^	0.64 ^3^	0.003 ^2^	0.35 ^3^	0.61 ^2^	0.27 ^3^
Potatoes and starchy foods, mushrooms (g/d)							
ICR	28.0 ± 30.5	43.5 ± 40.3	69.5 ± 26.5 *	36.5 ± 38.1	32.5 ± 25.6	40.9 ± 36.5	54.7 ± 23.9 *
CCR	31.5 ± 33.7	42.0 ± 38.9	42.3 ± 14.8 *	42.2 ± 35.0	16.7 ± 21.4	35.6 ± 30.4	−0.6 ± 30.1
*p*-value ^2,3^	0.59 ^2^	0.85 ^2^	0.66 ^3^	0.45 ^2^	0.79 ^3^	0.48 ^2^	0.35 ^3^
Milk, dairy products and cheese (g/d)							
ICR	197.9 ± 156.6	194.9 ± 141.9	9.1 ± 10.2	199.8 ± 143.2	12.8 ± 10.3	176.8 ± 153.8	−10.0 ± 10.2
CCR	187.5 ± 160.2	181.9 ± 96.6	18.0 ± 15.6	198.5 ± 121.4	14.6 ± 13.4	183.9 ± 119.7	18.5 ± 17.2
*p*-value ^2,3^	0.75 ^2^	0.61 ^2^	0.96 ^3^	0.97 ^2^	0.88 ^3^	0.82 ^2^	0.31 ^3^
Beef, veal, pork, mutton (g/d)							
ICR	11.9 ± 16.3	14.7 ± 16.2	−33.2 ± 28.2	14.2 ± 24.6	30.2 ± 30.8	14.3 ± 23.6	3.2 ± 23.4
CCR	21.4 ± 28.0	19.5 ± 21.4	−20.6 ± 21.6	13.0 ± 18.9	−12.3 ± 28.6	22.0 ± 26.6	−16.6 ± 18.3
*p*-value ^2,3^	0.043 ^2^	0.23 ^2^	0.35 ^3^	0.79 ^2^	0.10 ^3^	0.17 ^2^	0.57 ^3^
Game, poultry, offal (g/d)							
ICR	17.4 ± 21.9	21.8 ± 21.0	−1.1 ± 18.2	21.2 ± 25.8	−13.9 ± 22.1	15.3 ± 21.1	−10.9 ± 26.2
CCR	18.3 ± 23.4	17.0 ± 19.7	−10.2 ± 23.4	15.9 ± 17.2	−9.4 ± 18.8	16.6 ± 24.4	45.8 ± 19.0 *
*p*-value ^2,3^	0.84 ^2^	0.27 ^2^	0.29 ^3^	0.26 ^2^	0.28 ^3^	0.80 ^2^	0.91 ^3^
Sausage and processed meat (g/d)							
ICR	51.1 ± 44.4	26.0 ± 20.4	‑58.0 ± 15.4 *	23.6 ± 23.0	−64.9 ± 14.4 *	40.0 ± 33.1	−7.0 ± 13.8
CCR	31.5 ± 33.1	20.1 ± 24.7	−16.6 ± 20.2	21.5 ± 24.2	−24.8 ± 21.1	22.0 ± 19.1	−17.6 ± 19.3
*p*-value ^2,3^	0.015 ^2^	0.21 ^2^	0.08 ^3^	0.68 ^2^	0.016 ^3^	0.004 ^2^	0.79 ^3^
Fish and seafood (g/d)							
ICR	22.6 ± 40.3	22.3 ± 27.2	−38.7 ± 26.0	13.5 ± 17.1	−57.8 ± 23.8 *	11.5 ± 16.8	−46.3 ± 40.7
CCR	18.7 ± 24.6	17.9 ± 17.4	20.2 ± 25.4	16.9 ± 18.0	4.1 ± 28.3	16.9 ± 24.5	4.2 ± 26.0
*p*-value ^2,3^	0.57 ^2^	0.36 ^2^	0.97 ^3^	0.36 ^2^	0.25 ^3^	0.26 ^2^	0.27 ^3^
Eggs and egg products, pasta (g/d)							
ICR	41.0 ± 35.4	30.0 ± 27.0	−24.9 ± 16.9	32.4 ± 31.7	−18.0 ± 18.3	34.4 ± 37.7	−34.1 ± 21.0
CCR	52.5 ± 59.7	39.1 ± 35.8	−6.2 ± 19.1	34.9 ± 26.3	−22.7 ± 19.7	32.2 ± 25.5	−72.5 ± 28.2 *
*p*-value ^2,3^	0.25 ^2^	0.17 ^2^	0.88 ^3^	0.68 ^2^	0.47 ^3^	0.77 ^2^	0.25 ^3^
Fats and oils (g/d)							
ICR	18.1 ± 14.8	8.2 ± 5.8	−68.0 ± 14.3 *	9.3 ± 7.6	−58.3 ± 15.6 *	13.2 ± 11.8	−29.3 ± 18.5
CCR	17.3 ± 10.9	11.3 ± 6.4	−38.9 ± 13.3 *	11.0 ± 7.1	−35.6 ± 13.5 *	11.1 ± 8.5	−47.8 ± 13.8 *
*p*-value ^2,3^	0.76 ^2^	0.015 ^2^	0.11 ^3^	0.26 ^2^	0.33 ^3^	0.38 ^2^	0.71 ^3^
Legumes, nuts and seeds (g/d)							
ICR	9.2 ± 14.7	10.8 ± 17.0	−7.9 ± 31.4	12.6 ± 17.4	26.0 ± 31.7	6.7 ± 11.8	−20.4 ± 21.7
CCR	13.3 ± 15.8	20.7 ± 41.2	5.5 ± 32.1	12.6 ± 16.0	9.6 ± 30.9	15.5 ± 29.7	19.8 ± 36.5
*p*-value ^2,3^	0.19 ^2^	0.14 ^2^	0.42 ^3^	1.00 ^2^	0.35 ^3^	0.08 ^2^	0.38 ^3^
Bakery products, cakes and pastry (g/d)							
ICR	48.9 ± 41.5	18.4 ± 23.5	−79.7 ± 24.2 *	23.7 ± 25.5	−59.2 ± 18.3 *	46.2 ± 39.6	7.4 ± 15.2
CCR	52.7 ± 52.0	15.6 ± 17.5	−96.7 ± 23.2 *	25.5 ± 30.8	−53.4 ± 18.8 *	34.1 ± 27.5	−35.8 ± 20.7
*p*-value ^2,3^	0.68 ^2^	0.51 ^2^	0.45 ^3^	0.76 ^2^	0.72 ^3^	0.12 ^2^	0.07 ^3^
Sweets, sugar and ice cream (g/d)							
ICR	40.8 ± 37.2	19.6 ± 25.8	−78.3 ± 21.4 *	20.2 ± 20.4	−77.4 ± 19.8 *	25.7 ± 33.3	−86.7 ± 17.7 *
CCR	38.1 ± 34.8	14.3 ± 16.1	−92.5 ± 18.4 *	11.2 ± 11.2	−79.2 ± 23.9 *	26.7 ± 30.5	−37.6 ± 23.8
*p*-value ^2,3^	0.71 ^2^	0.24 ^2^	0.78 ^3^	0.010 ^2^	0.39 ^3^	0.88 ^2^	0.56 ^3^

^1^ Values are means ± SDs unless otherwise indicated. Changes are compared to baseline values and are shown as means of log_e_ percentage changes ± standard error of log percentage changes (%). ^*^ Significant within group change over time compared to baseline at *p* < 0.05 from paired t-test. Data were included from 98 participants (wk0 *n* = 49 for ICR and CCR; wk2 *n* = 45 for ICR, *n* = 46 for CCR; wk12 *n* = 47 for ICR (R days: *n* = 44), *n* = 46 for CCR; wk50 *n* = 41 for ICR (R days: *n* = 12), *n* = 38 for CCR). CCR, continuous calorie restriction; ICR, intermittent calorie restriction. ^2^
*p*-values for differences between groups at each time point were calculated by ANOVA *F*-test. ^3^
*p*-values for time-treatment interactions were calculated with linear mixed models adjusted for age and sex.

## Data Availability

Data described in the manuscript, code book, and analytic code will be made available upon request to the editors and reviewers by the corresponding author.

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
