# Peer review of "Adherence and Dietary Composition during Intermittent vs. Continuous Calorie Restriction: Follow-Up Data from a Randomized Controlled Trial in Adults with Overweight or Obesity"

_nutrients, 2021, doi:10.3390/nu13041195_

Round 1

Reviewer 1 Report

Thank you for the opportunity to review this manuscript.  It is well written and very comprehensive. I enjoyed reading this article and will use it as a model for how to explain methods, summarize data, and compare to other investigator's findings in the literature with my graduate students.  All of my questions were answered in the subsequent text and supplementary tables.  Strengths and limitations were appropriately addressed.  A few minor questions/suggestions:

  1. Possibly provide more detailed explanation as to why data was log transformed--you have done this but could be a teaching opportunity for graduate students reading this paper and needing to adjust their data to run statistical analyses.
  2. Appreciated that you evaluated data with respect to baseline isoenergetic energy requirement and actual baseline energy/nutrient intakes.  Wondered if there would be value in calculating/reporting/adjusting for isoenergetic energy requirement at each timepoint given change in weight--I know there are benefits and disadvantages to doing this.  Also wondered if Harris Benedict was the most appropriate formula to use with this sample instead of the St Jeor or other equations--you might want to address the reasoning for your choice of HB.
  3. In tables, some columns didn't include units of measurements--notice this in "Change" columns in particular--for instance in table 2, the change column should have had kcal/d as the unit.
  4. Table 3, seeing Fiber presented as E% is unusual as fiber contributes so little to energy intake.  Would be nice to see  an explanation of why you decided to do this and a brief explanation of types of fiber that contribute to energy intake.  Appreciated that Fiber in g/d is reported in supplemental tables.
  5. Would make more of your tables "landscape" orientation instead of "portrait" so that the font size can be increased.
  6. Appreciated and encouraged by weight loss that was achieved and maintained in this free-living, non-feeding intervention.  I know data is self-reported, but if it is true, adhering to the dietary recommendations provided to maintain a 5% weight loss for two years without additional intervention is hopeful.  Did you specifically assess whether participants were following any other type of weight loss intervention?  e.g., pharmaceutical, very low carbohydrate diet?  I know there were one or two follow-up questions about adherence to the assigned intervention and returning to less healthy dietary practices.  If so, you should mention these strategies as they could contribute to the weight loss maintenance that was achieved by both intervention groups.

Reviewer 2 Report

The connection between this article and the HELENA project should be explained much better in the abstract of the article. On a first reading of the abstract, it appears that the authors have performed a randomized trial again when this is not the case.

Judging the normality of a variable through a histogram does not seem adequate. The histogram depends on the origin and the bandwidth considered. I think that if the authors indicate that they use a normality test such as the Shpiro-Wilk test, I think it is superfluous to indicate that these visual techniques are also used (almost always ambiguous and doubtful) and especially when the sample size is not high.

The data presented in table 1 differ from the data in table 1 presented in the article "Effects of intermittent and continuous calorie restriction on body weight and metabolism over 50 wk: a randomized controlled trial" already published. The authors should review this table 1.

Figure 1 should be improved.

Reviewer 3 Report

Overall Comments

  • This is a well-written and interesting secondary analysis of the HELENA Trial. Most of my comments are minor suggestions with the exception of the authors’ use of an inaccurate method for predicting weight loss (i.e., the 3500 kcal rule).

Abstract

  • Please use people first language throughout the manuscript (e.g., people/participants with overweight or obesity rather than overweight and obese)
  • I realize space is limited in the abstract and the main goal is to compare diet composition/adherence between ICR and CCR, but can you squeeze in some information on the overall adherence? In other words, were there only limited differences between groups because neither were compliant or was compliance/adherence pretty good in both groups?

Introduction

  • Line 55-56: May be helpful to readers to provide some specific examples of “unfavorable food choices” that were reported in reference 20.

Materials and Methods

  • I appreciate the detailed explanation of the parent trial.

Results

  • There seems to be some broad systematic under-reporting of energy intake at all time points in the study. This is sufficiently addressed in discussion/study limitations.
  • Lines 202-203: The assumption that a 3500-kcal deficit leads to 1 pound of weight loss is a poor one that is not well-supported. I would advise against using this assumption and using alternative methods for determining expected weight loss.
  • See: https://www.ncbi.nlm.nih.gov/pmc/articles/PMC4035446/ and https://www.ncbi.nlm.nih.gov/pmc/articles/PMC3880593/
  • Pennington Biomedical Research Center and NIDDK both offer freely available and evidence-based weight loss calculators that the authors should consider for this analysis.
  • Figure 1 appears to be relatively low quality. Recommend a higher resolution figure. Also probably do not need to present weight change in the control since they are not otherwise included in these secondary analyses.

Discussion

  • Lines 317-320: This has the feeling of a major finding from this analysis that the authors may want to include in the abstract. This is sufficiently addressed later in the discussion. This reduction in adherence is clearly related to the greater average weight regain in ICR vs. CCR shown in Figure 1 and as reported in lines 258-261/Table S6 between those continuing to be adherent to the ICR protocol vs. those who were not. I feel as though this substantial difference in self-reported adherence to the ICR protocol indicates that ICR may be less sustainable for people in the long-term, which is counter to the hypothesis presented in the introduction (Lines 44-48).
